# Dimensions of Digital Literacy in the 21st Century Competency Frameworks

María Cristina Martínez-Bravo *[ID], Charo Sádaba Chalezquer [ID] and Javier Serrano-Puche [ID]

Department of Communication, University of Navarra, 31009 Pamplona, Spain; csadaba@unav.es (C.S.C.); jserrano@unav.es (J.S.-P.)
* Correspondence: mb.macristina@gmail.com or mb.martinez@gmail.com

**Abstract:** The Information and Knowledge Society demands the development of skills for the critical and responsible consumption and use of technology for leisure, personal, professional, academic development, and citizen participation. The international frameworks of "21st Century Competences" underline the importance of digital competence as the axis to enhance the rest of competences. This key competence goes beyond the operational use of technological tools and applications and has been studied from different approaches. This research explores digital literacy in eight international frameworks from different institutions and initiatives: UNESCO, European Union, OECD, ATCS, P21, NETS, NAEP, and Engauge, from which a content analysis is performed, and where the areas of scope of the competencies and the relationships between the different proposals are explored. The findings contribute to the understanding of an integrated approach to digital literacy, where six dimensions are identified: critical, cognitive, operational, social, emotional, and projective. Three dimensional profiles are also identified that point towards the critical use of technology, the appropriation of technology in daily life and social innovation, which invites a rethink towards digital literacy from a multi and interdimensional vision.

**Keywords:** 21st century skills; digital literacy; digital competence; future thinking; critical literacy; emotions; ICT; empowerment

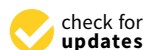



## 1. Introduction

The Internet has changed the notion we have of space, time, and the relationships between people and machines. From a spatial perspective, the Internet displays multiple digital ecosystems, virtual reality experiences, and storage in immaterial dimensions known as "clouds". Time has been marked by concepts such as ubiquity, synchrony and asynchrony of communications, and long-distance operations [1]. In the current human-machine interaction, the McLuhanist vision of "media as extensions of man" materializes thanks to technological development, where "interfaces not only converse with humans, but also converse with them" [2] and the ecological metaphor of media as environments, that is, the Internet as an ecosystem and a species in development [3], leads us to reflect on the perceptual and cognitive transformations of the human being in this environment, and learning in it. All this panorama leads to a reality where the Internet has changed the way in which knowledge, information, products, and services are produced, consumed, and exchanged; even the way of expressing emotions [4]. It has also made it possible to expand the public space for cultural, artistic, and social participation creation [5,6]. This dynamic accelerated by the technological revolution configures new relationships with the world, with society and culture, as well as a resignification of what is known.

At present, in the line of what is postulated by Bruno Latour [7] in his Theory of the Actor-network, which considers both humans and objects and discourses as actants, technology has an increasingly active role that comes from the "cerebralization" of the machine and inventions around artificial intelligence and the Internet of things. In other

words, technologies and people appear as inseparable elements. From a less symmetrical and more humanistic view, the relationship with technology proposed by Reig [8] implies the appropriation of these systems and tools to turn them into technologies for learning and knowledge (TAC) and technologies for empowerment and participation (TEP).

Participating in this movement and global system of knowledge and information requires the formation of a hybrid identity (connected and disconnected) and the construction of an analog and digital footprint in disruptive spatio-temporal scenarios. This challenge involves a cycle of lifelong learning and the development of new skills that allow adaptability to change and the complexity of systems. The so-called "21st century competences" or "Key Competences for Lifelong Learning" [9,10] have been explored by different entities and international organizations in order to promote the development of emerging skills not only technical, but also critical, cognitive and social.

The study by Voogt and Pareja-Roblin [11] points out that there are differences in the proposals regarding the categorization, grouping and hierarchization of these competencies in which, however, the centrality of digital competence is common [12] and covers various areas: critical, socio-cultural, communicational, creative, collaborative, technological, operational, and responsive.

The research by Van-Laar et al. [13] on the relationship between 21st century competencies and digital competences identifies 12 skills that are divided into core (technique, information management, communication, collaboration, creativity, critical thinking, and resolution of problems), and contextual (ethical awareness, cultural awareness, flexibility, self-direction, and lifelong learning). For their part, other proposals also include approaches to adaptability to change, the development of future thinking [14], the management of complexity, creative thinking, or digital literacy with a scientific, technological, and economic base in a multicultural environment [15].

Along with the clear potential of the Internet for information acquisition and cognitive development, the Internet is also an affective technology, in the sense that it is a vehicle for the expression of emotions [16,17]. Furthermore, the Internet not only arouses emotions in its users and serves as a channel for the expression of affections, but also influences the way in which this affection is modulated and deployed, as well as the configuration of a person's identity. Given the affective charge associated with the use of technology, digital habits for acquiring information and being in contact with other users always have an emotional component. This is characterized by its complexity, as it encompasses both positive feelings (increased connectivity with family and acquaintances, sense of belonging to a social group, etc.) and negative feelings (emotional dependence on the cell phone, anxiety or information saturation, etc.) [18].

All this leads to highlight the value of the emotional dimension in the acquisition of digital competence, and how it must be integrated into a holistic view of the Internet and the literacy it requires. Faced with the "reading and writing" of classical literacy, this digital culture now calls for the development of different competencies and abilities, both instrumental and cognitive-intellectual, sociocultural, axiological, and emotional [19]. Therefore, it is advisable to be aware of the risk of "reducing digital competence to its most technological and instrumental dimension: focusing on technical knowledge, on procedures for the use and management of devices and programs, forgetting attitudes and values" [20] (p. 31). On the contrary, it is essential to articulate the socio-emotional level in the set of dimensions of digital competence.

Previous studies have associated digital competencies to different dimensions [21–23], for example, the study by Area-Moreira [24] identifies five dimensions: the instrumental, related to the technical domain; the cognitive, related to the acquisition of new knowledge and skills; the communicative, focused on communication and personal interactions; the axiological, aimed at the development of ethical, democratic, and critical values towards technology; and the emotional dimension, related to the set of affections, feelings and emotions that arise in the experience in digital ecosystems. Thus, the instrumental nature of technology has been nurtured from multiple perspectives and has reached a critical and

reflective approach [25] oriented to techno-social empowerment [26], for the construction of a digital citizenship [27].

In short, the development of digital competence favors the safe, critical, and creative use of ICTs for employment, learning, leisure, personal development, and participation in society [28]. This literacy has been integrated and recognized in the agenda of several international organizations, and, consequently, countries have renewed their educational policy and legislation, as well as the education ecosystem [29]. The implementation of 21st century competencies and, in particular, digital competence in educational public policy, implies the execution of a well-articulated plan [30].

The present research explores the dimensions of digital literacy in eight international frameworks or frameworks of 21st century competencies from different institutions and initiatives: ATCS, Engauge, NAEP, NETS, OECD, P21, UNESCO, and European Union (EU). Based on the hypothesis that the different approaches with which digital competence has been developed in these frameworks are interrelated, and that their interdisciplinarity enriches the construction of an integrated proposal, the objective is to identify through a content analysis the dimensions of digital literacy developed in the different proposals, as well as the dimensional approach of each framework, in order to contribute with a more enriched perspective of digital literacy.

## 2. Research Methodology

*Selection of Cases and Analysis Unit*

Based on previous research by Voogt and Pareja-Roblin [11], eight proposals for competency frameworks were selected, as they are considered significant contributions, most of which have been periodically updated. In addition, it is important to highlight that these contributions were also selected because, according to the same study, within them, digital competence stands out as core competence, which allows us to delve into its dimensions. The eight proposals that were analyzed in their updated versions are: (1) ATCS, (2) Engauge, (3) NAEP, (4) NETS, (5) OECD, (6) P21, (7) UNESCO, and (8) European Union (Table 1).

**Table 1.** Twenty-first century competency frameworks analyzed (cut-off date: August 2020).

| N° | Base Document Name | Acronym | Organization/Entity | Scope | N° Documents Reviewed | Literacy Focus |
|---|---|---|---|---|---|---|
| 1 | Assessment and Teaching of 21 Century Skills | ATCS | International Project sponsored by Cisco, Intel y Microsoft. | International | 9 | Computer and Information Literacy Digital Literacy |
| 2 | enGauge 21st Century Skills: Literacy in the Digital Age | enGauge | North Central Regional Educational Laboratory (NCREL) and Metiri Group. Document produced with funds from the US Department of Education. | National | 1 | Literacy in digital age |
| 3 | Technological Literacy and Engineering Framework for 2018. National Assessment of Educational progress. | NAEP | Developed by WesEd, requested by the US Government | National | 4 | Technology and Engineering Literacy |
| 4 | National Educational Technology Standards | NETS | International Society for Technology in Education (ISTE) | National | 12 | Literacy in digital age |
| 5 | OECD Future of Education and Skills 2030 | OECD | Organisation for Economic Co-operation and Development | Regional | 23 | Digital literacy |

**Table 1.** *Cont.*

| N° | Base Document Name | Acronym | Organization/Entity | Scope | N° Documents Reviewed | Literacy Focus |
|---|---|---|---|---|---|---|
| 6 | Partnership for 21st century skills | P21 | US government and private organizations (Apple Computer Inc., Cisco Systems, Dell Computer Corporation, National Education Asociation, among other.) | International | 7 | Information literacy Media literacy ICT literacy |
| 7 | A Global Framework of Reference on Digital Literacy Skills for Indicator 4.4.2 | Unesco | United Nations Educational, Scientific and Cultural Organization, UNESCO | International | 6 | Digital literacy |
| 8 | Digital Competence for lifelong Learning | EU | European Union | Regional | 8 | Digital literacy |

For the selection of the unit of analysis, an exploratory review was made of seventy main and procedural documents, linked to the frameworks, from which a database of digital competencies is constructed and of the definitions set out in the documents, elements that constitute the basis of the analysis.

In order to study the consistency and focus of the dimensions of digital literacy in the different framework proposals, a content analysis is applied. For the procedure of the qualitative content analysis technique, the guidelines of Mayring, cited by Cáceres [31], are followed, with which the process is developed in five stages, summarized in Table 2. The content analysis is performed with the open access software QCAMap.

**Table 2.** Content analysis process applied in the study.

| N° | Stage | Description of Activities/Stage | Application |
|---|---|---|---|
| 1 | Selection of a communication model | Scope | Analysis of digital competencies in the context of the 21st century competency frameworks |
| 2 | Pre-analysis | First level of information organization | 1. Selection of eight sources of 21st Century competency framework documents: UNESCO, European Union, ATCS, P21, NETS, NAEP, Engauge, and OECD.<br>2. Identification and verification of the digital competencies section in the reviewed documents.<br>3. Exploration of pre-analysis categories, which are later validated in the coding stage. |
| 3 | Definition of analysis unit | Identification of the body of text to be analyzed | Development of a database with paragraphs containing digital skills and their definitions by analyzed framework. |
| 4 | Establishment of analysis rules and classification codes. | Definition of categorization system | Six categories and their descriptive context are established, a summary below:<br>1. Operational dimension: technological, functional, and instrumental area.<br>2. Social dimension: area of communication, participation, collaboration, exchange, and culture.<br>3. Emotional dimension: personal area, empowerment, identity, and emotional relationship.<br>4. Cognitive dimension: higher order competence area. Creativity, production, problem solving, etc.<br>5. Critical dimension: decisions, values, and attitudes towards situations and contexts.<br>6. Projective dimension: innovation, future thinking, scenario building, systemic thinking, theorizing, and inventiveness. |
| 5 | Synthesis | Interpretation and extrapolation | The QCAMap software was used to categorize and create a coded database, which was used to process the material for analysis, synthesis and subsequent definition of the categories identified. The interpretation of the data was also complemented with the Orange software. |

To complement the processing and interpretation of the data obtained from the content analysis, the free data mining software Orange was used, with which a principal component analysis (PCA) was applied to reduce the variables according to their behavior, including 76% of the variability, that is, the representativeness of the data analyzed.

This process shows the determining characteristics of the differences in the frameworks. Dimensional profiles of the frameworks were also constructed, expressed in clusters with the application of the Hierarchical Clustering method, with which the frameworks were grouped according to the similarity in the behavior of their variables (dimensions). For data analysis and systematization, it was considered that the frequency of the analyzed code (body text) was heterogeneous and influenced by the size of the analyzed text, so it was necessary to normalize the values on a percentage scale.

## 3. Results

### 3.1. Dimensions of Digital Literacy in the Frameworks of 21st Century Competencies

The identification of dimensions, understood in this study as large areas where the competencies are inherent to digital literacy converge, implies a metacognitive exercise of articulation of the different proposals. In these dimensions, in a second phase of research, the different skills, abilities, and attitudes will be grouped.

From the content analysis of the competencies and their definitions, it has been possible to identify six dimensions of digital literacy: (1) critical, (2) cognitive, (3) social, (4) operative, (5) emotional, and (6) projective (Figure 1), which will allow the integrated organization of the contributions of the various proposals.

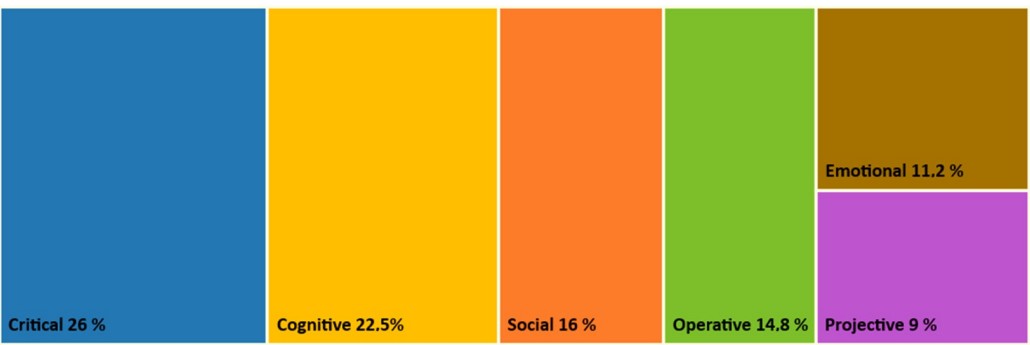

**Figure 1.** Dimensions of digital literacy identified in international reference frameworks.

According to the recurrence of the code identified in the analyzed frameworks, the most developed dimensions are the critical (26%) and cognitive (22.5%) ones. This allows us to assess that the instrumental vision of digital literacy has taken a back seat. According to the findings, each dimension, including the operational one, implies the development of skills connected to needs and based on a sense of integral purpose, as presented below in the description of each of the dimensions identified:

### 3.1.1. Critical Dimension

This implies adopting a position, attitude, and values when faced with diverse situations, cultivating social and civic responsibility, as well as developing the judgment to assess and make effective decisions in the face of risks, and to develop attitudes of self-control, autonomy, and flexibility. Participation and critical use of information, tools and collaborative spaces and media are essential, as is the fundamental understanding of the ethical and legal issues surrounding the digital ecosystem, understanding the messages behind the ideas that are dispersed in multiple media, and paying special attention to the misuse of information. In this area are issues such as security, digital standards and rights, intellectual property, and environmental protection. In other words, here we find competencies that make sense of the tools and build meaningful connections.

### 3.1.2. Cognitive Dimension

This is a dimension that integrates high-level competencies such as problem solving, management of complexity or complex environments, development of logical reasoning, cognitive processes of analysis, comparison, inference, interpretation, evaluation, creativity, and production. Planning and results management are also essential elements, understood as the ability to organize oneself to efficiently achieve objectives. In the digital environment it involves the design and development of systems; the understanding of scientific concepts and processes; the processes of creation; the curation of a variety of resources using digital tools to build knowledge, produce creative artifacts, and create meaningful learning experiences, linked to personal learning objectives; and the development of strategies that take advantage of technology to achieve them and the reflection on the learning process to improve its results.

### 3.1.3. Social Dimension

In this dimension the development of the sense of belonging to a global community, the multicultural vision, participation in networks and communication in the digital ecosystem constitute the starting point. Teamwork and collaboration, together with leadership skills, generate opportunities for exchange between two or more people to connect needs, motivations, solve problems or to create new products/ideas. This dimension also implies the development of digital citizenship, the search for opportunities for self-development and empowerment in the use of technologies. The skills to live, learn, and work in an interconnected digital world, enriched by collaboration with others locally and globally are essential.

### 3.1.4. Operative Dimension

This involves problem solving from a more instrumental/technological scope; the ability to use tools, the exchange, interaction, and execution of tasks adapting to the dynamic nature of digital environments and their protocols. It also implies an understanding and resolution of possible technical problems, understanding of programming principles, data manipulation, software and hardware operation, and configuration and modification of programs and devices. Of additional importance are the processes of technology evaluation (applicable to other environments) and the understanding of the design process, for the development, testing, and refinement of prototypes as part of a cyclical design. In this dimension, ICTs reach their maximum meaning when effective use is made of tools connected to real-world needs.

### 3.1.5. Emotional Dimension

This dimension includes the management of emotions, one's own behavior, and the construction of healthy relationships. Therefore, it involves skills that imply the ability to read, manage emotions, motivations, and behaviors of oneself and others during a social interaction. These include the development of interpersonal skills for exchange and collaboration, the management of digital identity, the sense of protection of one's own humanity against the risks that the Internet can represent, both physically and psychologically, as well as the cultivation of curiosity as the human desire/drive to learn. All this integrated with the feeling of being connected to personal and/or social human needs and motivations.

### 3.1.6. Projective Dimension

This dimension implies the recognition and awareness of living in complex and dynamically changing environments and situations. It involves acquiring knowledge to make predictions and solve problems based on innovative technologies; the development of the capacity for innovation, the capacity for inventiveness, future thinking, computational thinking, algorithmic thinking, the capacity to recognize patterns, modeling, and data management; the capacity to theorize, to test ideas and theories, to model processes; and the capacity to modify thinking, attitude or behavior to better adapt to the current or future

environment, while being aware of the limitations of time, resources and systems. In short, the ability to build scenarios with all this knowledge base.

### 3.2. Three Dimensional Profiles and Approaches to Frameworks

Another of the study's findings is the profile of the frameworks studied according to the dimensions developed. The application of the principal component analysis (PCA) allowed the main characteristics of the differences of the frameworks to be visible. In turn, the data were processed through the hierarchical clustering tool in which three profiles or clusters of frameworks were identified (Figure 2).

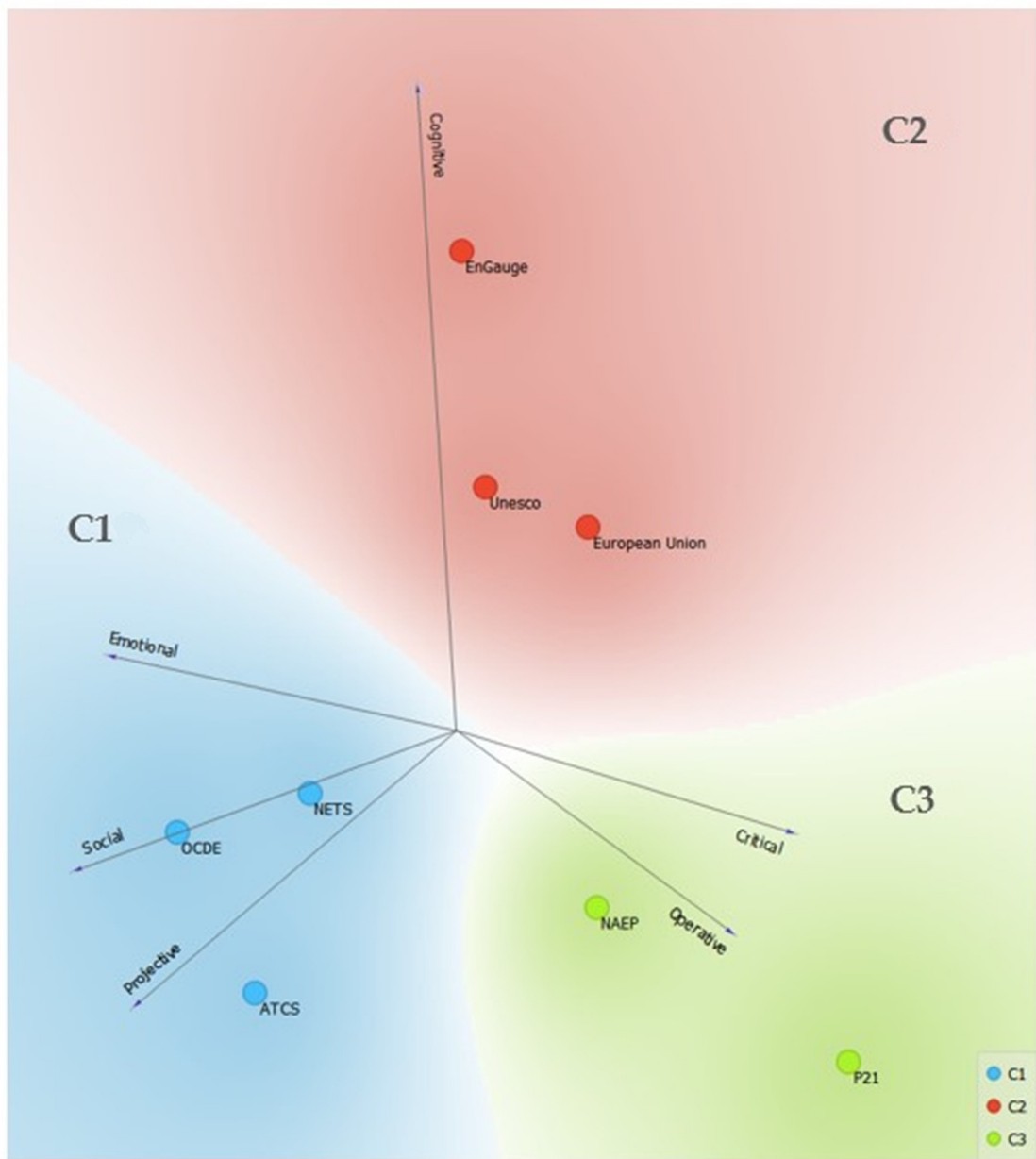

**Figure 2.** Profiles and dimensional approach of frameworks.

The projective socio-emotional profile (C1 = cluster 1, blue color) integrates the OECD, ATCS, and NETS frameworks, and develops several dimensions with a strong emphasis on personal and community development. In the cognitive profile (C2 = cluster 2, red color) are the Engauge, UNESCO, and European Union frameworks with an emphasis on the development of logical or higher order skills, which, in turn, are connected to the critical

and operational approach. Finally, the operational-critical profile (C3 = cluster 3, green color) places two frameworks, distanced from each other, NAEP and P21. They mainly develop the operational dimension with a critical approach.

The location of each dimension is plotted along an axis; the central point of the axes corresponds to the average. That is, the closer the points with the names of the frameworks get to the center, the less specificity in one dimension and the less distance from the average. Although frameworks favor some approach in one or more dimensions (Figure 3), most of the proposals develop all the dimensions presented in the previous section, although in different proportions.

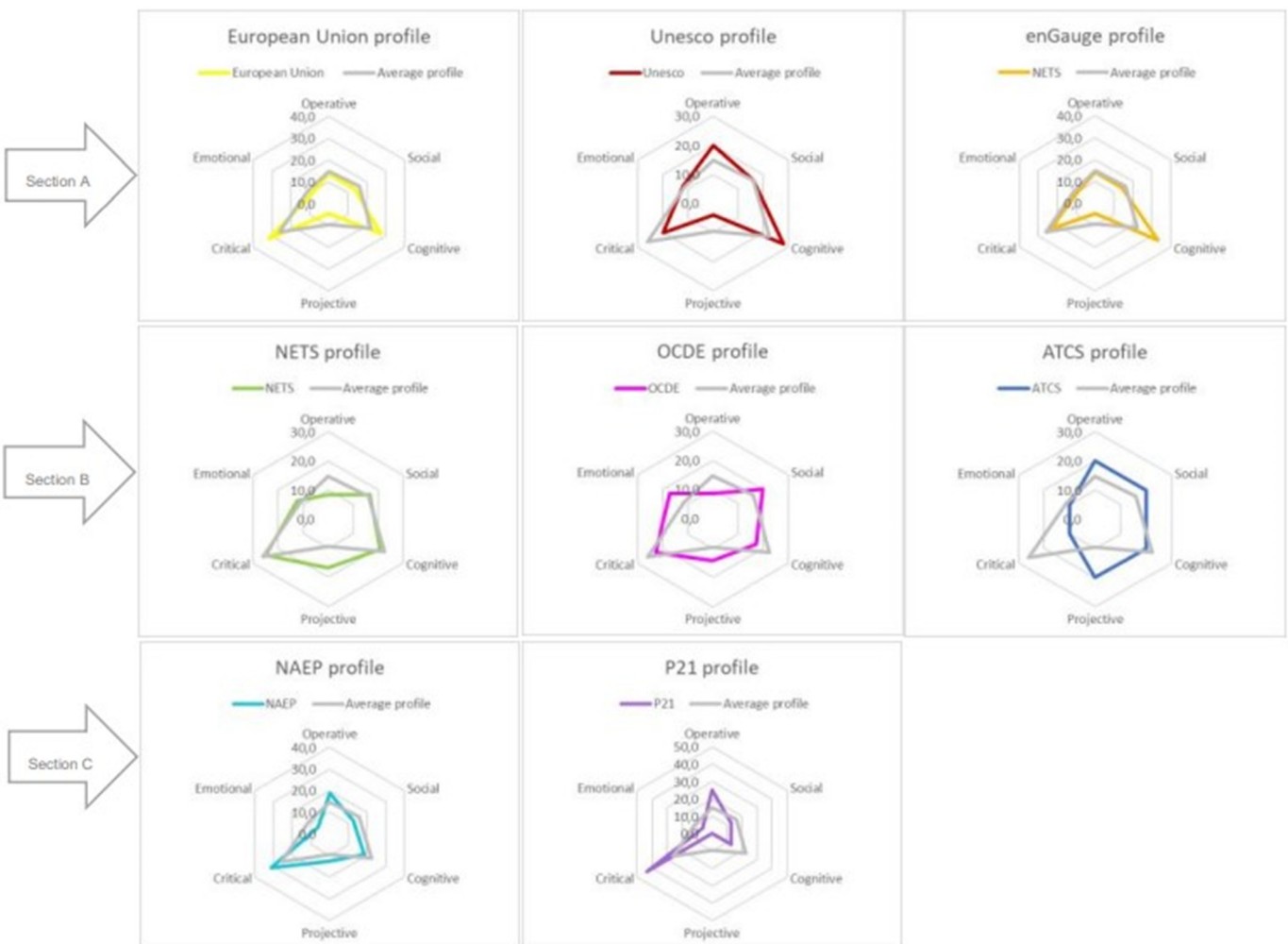

**Figure 3.** Comparison of the average profile versus the profile of each analyzed framework.

Figure 3 shows independently the specialization of each frame of reference. These have been organized according to the clusters described above. In the graph, each profile is differentiated by a color and the average profile is marked in gray.

3.2.1. Technology for Empowerment in Everyday Life

Section A contains the frameworks of the European Union, UNESCO, and Engauge, which are more homogeneous among themselves. These frameworks develop mainly the cognitive dimension, followed by the critical and operational dimension. That is, there is a vision of the use of technology with a logical, critical, and technical approach, in other words, a creative, ethical, and practical use of technology, aimed at solving daily situations. It is a use thought at the community or relationship level (social life, work, and citizen participation), less linked to the development of the person and their emotions.

### 3.2.2. Technology for Social Innovation

Section B shows a second group of frameworks where the NETS, OECD, and ATCS proposals are found, which are more heterogeneous among themselves, but are marked by three dimensions: social, projective, and emotional. Therefore, it is an approach towards a socio-emotional technology; here the projective plane is shaped towards social innovation, where logic, technique, and criticism have a humanistic approach. It is important to note that the three proposals develop all dimensions in a better distributed way. Therefore, we are talking about more integral proposals.

### 3.2.3. Critical Technology: Building the Techno-Social Vision

Finally, section C groups two frameworks: NAEP and P21. These frameworks are different from one another, but with a strong emphasis on critical technology; that is, a use where the context (natural, social, political, citizen, etc.) plays a fundamental role, as well as ethical and aesthetic values. In other words, the constructivist approach to technology, responsible autonomy, and values are essential for the meaning that technology acquires in society and in people. In the case of the NAEP proposal, all dimensions are developed in a more distributed manner. On the other hand, the P21 profile does not develop the projective dimension.

### 3.2.4. Multidimensional and Interdimensional Digital Literacy

The data in Figure 3 are complemented with Table 3, which shows how the dimensions are distributed in percentage terms for each framework. It can be seen that there are proposals in which a certain dimension is privileged. For example, in the P21, European Union, and NAEP proposals, the critical dimension carries the greatest weight. On the contrary, the least developed dimension in all the proposals is the projective one; however, the ATCS, NETS, and OECD proposals have the highest values in this dimension. This also highlights a great development of the emotional dimension in the OECD framework.

**Table 3.** Percentage contribution of dimensions identified in each framework.

| Frameworks | Dimensions | | | | | | |
|---|---|---|---|---|---|---|---|
| | Operative | Social | Cognitive | Projective | Critical | Emotional | Total |
| ATCS | 20% | 20% | 20% | 20% | 10% | 10% | 100% |
| EnGauge | 14% | 14% | 33% | 5% | 24% | 10% | 100% |
| European Union | 14% | 14% | 27% | 5% | 32% | 9% | 100% |
| NAEP | 19% | 13% | 19% | 13% | 31% | 6% | 100% |
| P21 | 25% | 13% | 13% | 0% | 44% | 6% | 100% |
| NETS | 8% | 17% | 21% | 17% | 25% | 13% | 100% |
| Unesco | 20% | 16% | 28% | 4% | 20% | 12% | 100% |
| OECD | 9% | 20% | 17% | 14% | 23% | 17% | 100% |
| Average profile | 14.8% | 16.0% | 22.5% | 9.5% | 26.0% | 11.2% | 100% |

In some proposals, the dimensions are distributed in a more uniform way, which implies a commitment to a multidimensional and interdimensional digital literacy that seeks techno-social empowerment, based on the development of various dimensions that complement each other, and that require each other, for the use of technology in different contexts, which implies an interdimensional relationship.

It is important to highlight that, despite the differences, there is a solid agreement in the commitment to the critical dimension of digital literacy, which allows the development of an information and knowledge society that pursues meaningful connections, within a framework of constructive attitudes, responsibilities, and values.

## 4. Discussion

Education faces global challenges in the face of volatile, uncertain, complex, and ambiguous conditions of a "VUCA world", susceptible to deep reflection [32] (pp. 40–41), as well as challenges in the face of a global citizenship [27] and its non-existent "universal state" [33], which is inserted without structure in this "global village" hand in hand with the technological revolution and innovation. This scenario implies the development of competencies for coexistence on a global scale in increasingly challenging ecosystems.

Consistent with the context, the frameworks analyzed express a consensus towards a technocritical approach, which goes beyond the operational vision of technologies [34,35]. Although for some perspectives, critical thinking is located in the cognitive area; since it is conceived as a complex cognitive process [36], beyond the taxonomic structure, the current panorama of studies that decant for this vision denote that the relevance that this educational dimension has acquired responds to that prevailing and emerging need for ethical and reflective conditions required in the world mediated by technologies [37]. A world where algorithms build scenarios, in which human beings give meaning to their lives in different contexts, without being certain of their own autonomy, in an entangled and veiled technocratic relationship of "impersonal domination" [38] (pp. 6–8).

And it is precisely in the sense that technology acquires where digital literacy, proposed in the competency frameworks of the 21st century, supports the appropriation of skills and techno-social empowerment, since even in the most instrumental dimension, it requires a context and purpose that connects the tools to the needs of the real world [28,39]. In this same line, from the critical theory of technology proposed by Feenberg [38], it is considered that, in essence, technology contains two aspects, a primary and a secondary instrumentalization, which are explained from the theory of instrumentalization. This theory proposes that technology should be analyzed at two levels; in the first there is a process of "de-worlding" (decontextualizing it to leave it at the functional level), and in the second, it is "worlded" again (contextualizing it in the natural or social environment). In other words, the human condition (subjective) cannot be disconnected from the technical condition (objective) [40]. This critical approach converges towards the democratization of technology for social appropriation, democratization that has been limited by multiple structural factors [41] (pp. 198–199).

Although the human conditions of technology and critical literacy are not new perspectives of study, there is a flourishing of some approaches such as the concept of "purpose" in education, which emphasizes the purpose of learning and the meaning it gives and has in the life of the person [32].

Both from the critical theory and from the humanist and constructivist visions of technology, the critical use of technologies has to do with attitudes and the ability to deal with various situations at the personal, social, cultural, civic level, etc [42]. That is, put things in context so that they make sense, and, consequently, define a position, decision, or action.

For its part, the emotional dimension of digital literacy also refreshes classic discussions on the freedoms of the individual, human behavior, morality, ethics and aesthetics [43], and the condition of the citizen facing a "bigger" and hyperconnected world, developed in the perspective of digital humanities. This scenario demands the need for another type of leadership for liquid times [44], which overcome the efficiency-based vision of education and the development of competencies [45].

In line with the projective dimension identified in this study, it is worth mentioning the work that UNESCO has been developing since 2012 regarding futures literacy, which "empowers the imagination, enhances our ability to prepare, recover and invent as changes occur" [46]. Thus, this approach, and in general the future studies perspective [47], is reflected in the projective dimension of digital literacy, both in the anticipatory, reflective, inventive and/or innovative capacity, as applied in algorithmic thinking, appropriation and modeling of data, the ability to theorize, among others.

In this sense, future research oriented to the exploration of prospective scenarios for democratization and appropriation are essential in order to develop the humanistic perspective and the development of an Internet governance ecosystem aimed at social empowerment and the multidimensional approach to digital literacy.

## 5. Conclusions

The six dimensions of digital literacy identified in the eight international framework proposals integrate a holistic vision of the Internet and the possibility of building an information and knowledge society that empowers people's lives. Faced with this, there remains the great challenge of making this literacy a reality on a large scale and making the democratization of knowledge and access to technology a pillar of public education policy.

In the study, seven of the eight proposals of reference frameworks analyzed (EnGauge, NAEP, European Union, NETS, UNESCO, and OECD) develop, although in different profiles, all the dimensions identified: critical, cognitive, operative, social, emotional, and projective. While only one of the proposals analyzed (P21) does not contain one of these dimensions.

The emphasis on the critical dimension of digital literacy in the analyzed reference frameworks is revealingly highlighted, which constitutes a great commitment to the construction of significant ecosystems and the development of an awareness and values connected with social and civic responsibility in a globalized world.

The cognitive dimension is also of great importance in the proposals, derived from schemes of complex thinking and problem solving, coherent in a world increasingly full of economic, social, cultural, etc., challenges.

A growing projective approach is also identified, derived from the, so-called, future studies, where technological tools are used to build scenarios that make it possible to project the development of people's own lives and the functioning of systems.

Finally, it is important to point out that the different proposals allow the construction of an integral perspective, which leads us to rethink digital literacy from a multi and interdimensional vision. This implies a commitment to techno-social empowerment with a humanistic approach, aiming at social innovation, critical and autonomous use of technology, and creative, reflective, and responsible appropriation of it in everyday life.

**Author Contributions:** Conceptualization, M.C.M.-B., C.S.C. and J.S.-P.; methodology, M.C.M.-B.; software, M.C.M.-B.; validation, C.S.C. and J.S.-P.; investigation, M.C.M.-B.; writing—original draft preparation, M.C.M.-B.; writing—review and editing, C.S.C. and J.S.-P.; visualization, M.C.M.-B.; supervision, C.S.C. and J.S.-P.; project administration, C.S.C.; funding acquisition, C.S.C. and J.S.-P. All authors have read and agreed to the published version of the manuscript.

**Funding:** This research received no external funding.

**Institutional Review Board Statement:** Not applicable.

**Informed Consent Statement:** Not applicable.

**Data Availability Statement:** Not applicable.

**Conflicts of Interest:** The authors declare no conflict of interest.

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
