# Peer review of "Dimensions of Digital Literacy in the 21st Century Competency Frameworks"

_sustainability, doi:10.3390/su14031867_

Round 1
Reviewer 1 Report
It is a valuable paper with a survey of current digital literacy research in eight international frameworks.
Author Response
Dear reviewer,
We are very grateful for the comment.
Best regards
Reviewer 2 Report
It is adequately structured research, both in the methodological field and in the construction of the report. I have only one observation for future improvements, regarding section 3.2: the principal component analysis (PCA) to make visible the main characteristics of the differences of the frameworks, could be improved from other techniques for small data, such as k-means, and thus to improve the construction of the clusters.
Congratulations for the excellent work done. It will be very useful to build better educational proposals.
Author Response
RESPONT REPORT
Comments and Suggestions for Authors
Reviewer
It is adequately structured research, both in the methodological field and in the construction of the report. I have only one observation for future improvements, regarding section 3.2: the principal component analysis (PCA) to make visible the main characteristics of the differences of the frameworks, could be improved from other techniques for small data, such as k-means, and thus to improve the construction of the clusters.
Congratulations for the excellent work done. It will be very useful to build better educational proposals.
Authors
Dear reviewer,
We are very grateful for the comments and for nurturing our research. We also take this opportunity to comment that the data were subjected to multiple clustering methodologies, including k-means. However, the hierarchical clustering method allowed us to establish the number of clusters by visual analysis of the dendogram. In contrast, the k-means method required the establishment of a number of clusters to run the analysis, which is more applicable when there are a number of pre-established categories, in our case we did not have such pre-defined categories.
It is worth mentioning that, however, when applying the k-means method the results in the composition of the clusters did not change (I attach the data set of that procedure).
Thanks again for the valuable input and we look forward to implementing this and other recommendations for future research.
Best regards,
